



**Retrieval and analysis of the composition of an aerosol mixture through Mie-Raman-**
**Fluorescence lidar observations.**
Igor Veselovskii[1], Boris Barchunov[1], Qiaoyun Hu[2], Philippe Goloub[2], Thierry Podvin[2], Mikhail
Korenskii[1], Gaël Dubois[2], William Boissiere[2], Nikita Kasianik[1]
*[1]Prokhorov General Physics Institute of the Russian Academy of Sciences, Moscow, Russia.*
*[2]Univ. Lille, CNRS, UMR 8518 - LOA - Laboratoire d'Optique Atmosphérique, F-59650 Lille,*
*France*
**Correspondence**: Philippe Goloub (philippe.goloub@univ-lille.fr)
**Abstract**
In the atmosphere, aerosols can originate from numerous sources, leading to the mixing of different
particle types. This paper introduces an approach to the partitioning of aerosol mixtures in terms
of backscattering coefficients. The method utilizes data collected from the Mie-Raman-
fluorescence lidar, with the primary input information being the aerosol backscattering coefficient,
particle depolarization ratio ($\delta$), and fluorescence capacity ($G_F$). The fluorescence capacity is
defined as the ratio of the fluorescence backscattering coefficient to the particle backscattering
coefficient at the laser wavelength. By solving a system of equations that model these three
properties ($\beta_F$, $\delta$ and $G_F$), it is possible to characterize a three-component aerosol mixture.
Specifically, the paper assesses the contributions of smoke, urban, and dust aerosols to the overall
backscattering coefficient at 532 nm. It is important to note that aerosol properties ($\delta$ and $G_F$) may
exhibit variations even within a specified aerosol type. To estimate the associated uncertainty, we
employ the Monte Carlo technique, which assumes that $G_F$ and $\delta$ are random values uniformly
distributed within predefined intervals. In each Monte Carlo run, a solution is obtained. Rather
than relying on a singular solution, an average is computed across the whole set of solutions, and
their dispersion serves as a metric for method uncertainty. This methodology was tested using
observations conducted at the ATOLL observatory, Laboratoire d'Optique Atmosphérique,
University of Lille, France.
**1. Introduction**



Studying the physicochemical properties of atmospheric aerosols is crucial for
understanding their impact on Earth's radiation balance and climate. To simplify the complexity
of aerosol composition, it is essential to classify aerosol types. Categorization of aerosols into
several basic types, e.g. urban, dust, marine, biomass burning (Dubovik et al., 2002), allows to
cover the range of variability of observed aerosol parameters and facilitates the analysis and
interpretation of aerosol data. The multiwavelength Mie-Raman and HSRL (High Spectral
Resolution Lidar) lidar systems provide an unique opportunity to derive height-resolved particle
intensive properties, such as Angstrom exponents, lidar ratios, and depolarization ratios at multiple
wavelengths. These properties can be used as inputs for classification schemes (Burton et al., 2012,
2013; Groß et al., 2013; Mamouri et al., 2017; Papagiannopoulos et al., 2018; Nicolae et al., 2018;
Hara et al., 2018; Voudouri et al., 2019; Wang et al., 2021; Mylonaki et al., 2021; Wandinger et
al., 2023; Floutsi et al., 2023b). However, aerosols in the atmosphere often originate from multiple
sources, leading to the mixing of different particle types. To understand the impact of different
aerosol types within a mixture, it is necessary to quantify the content of each type.
In the cases involving mixtures of two aerosol types with significantly different
depolarization ratios, the partitioning of aerosol backscattering coefficients becomes
straightforward (Sugimoto and Lee, 2006; Tesche et al., 2009; Miffre et al., 2020). Burton et al.
(2014) have formulated the mixing rules for several aerosol intensive parameters, such as lidar
ratio, backscatter color ratio, depolarization ratio, and applied these rules to two-component
aerosol mixtures. However, the partition becomes increasingly challenging when dealing with
more than two types of particles. The limited number of lidar-measured intensive particle
properties specific to individual aerosol types contributes to this challenge. Even for a single
aerosol type, the measured particle parameters, such as lidar ratios, demonstrate a wide range of
variability (Floutsi et al., 2023a). Distinguishing between urban and smoke particles poses a
particular challenge as these two types exhibit similar lidar-measured properties (Floutsi et al.,
2023a). Therefore, additional independent information is needed to enhance the characterization
of aerosol parameters.
Independent information about aerosol properties can be obtained through fluorescence lidar
measurements (Reichardt et al., 2018, 2023; Veselovskii et al., 2020; Zhang et al., 2021). The
fluorescence lidar allows evaluating the fluorescence backscattering coefficient $\beta_F$, which is
derived from the ratio of fluorescence and nitrogen Raman backscatters (Veselovskii et al., 2020).



The particle intensive property in fluorescence lidar measurements is the fluorescence capacity $G_F$, which is the ratio of $\beta_F$ to the aerosol backscattering coefficient at the laser wavelength. The fluorescence capacity of smoke is approximately one order higher than that of urban particles, providing a basis for distinguishing between these two aerosol types (Veselovskii et al., 2022). Additionally, recent studies have shown that a classification scheme relying on two intensive parameters - the particle depolarization ratio at 532 nm ($\delta_{532}$) and the fluorescence capacity, effectively separates four aerosol types: dust, smoke, pollen, and urban (Veselovskii et al., 2022). It is noteworthy that the current classification scheme does not discriminate particles based on their absorption properties, so the "urban" type encompasses both continental aerosol and anthropogenic pollution. Furthermore, maritime aerosol is not included in the classification at present, as the lidar observations were performed over Lille, where maritime particles are not prevalent (though the possibility of its inclusion is acknowledged).

In this study, we extended the approach beyond classification to partition aerosol mixtures in terms of the backscattering coefficients of basic aerosol types. To test the approach, we analyzed observations at the ATOLL (ATmospheric Observation at liLLe) at Laboratoire d'Optique Atmosphérique, University of Lille, between 2020 and 2023, performed during periods of strong smoke and dust episodes. We begin by providing a description of the lidar system and the approach for mixture partition in Section 2. In the first part of the results section (Section 3.1), we present two case studies that demonstrate how the algorithm operates. In the second part (Section 3.2), we analyze the results obtained during the heatwave in July 2022. The paper concludes with a summary of our findings in the conclusion section.

## 2. Experimental setup and approach to aerosol mixture partition

### 2.1. Lidar system.

The Mie-Raman-fluorescence lidar LILAS (LIlle Lidar AtmosphereS) is equipped with a tripled Nd:YAG laser that operates at a repetition rate of 20 Hz and has a pulse energy of approximately 100 mJ at 355 nm. A 40 cm aperture Newtonian telescope is utilized to collect the backscattered light, and Licel transient recorders with a range resolution of 7.5 m are employed to digitize the lidar signals. This configuration allows for simultaneous detection in both analog and photon counting modes. The objective of the LILAS system is to detect elastic and Raman backscattering, which enables the measurement of various properties through the 3β+2α+3δ data





configuration. This includes three particle backscattering coefficients ($\beta_{355}$, $\beta_{532}$, $\beta_{1064}$), two
extinction coefficients ($\alpha_{355}$, $\alpha_{532}$), and three particle depolarization ratios ($\delta_{355}$, $\delta_{532}$, $\delta_{1064}$). The
particle depolarization ratio, determined as a ratio of cross- and co-polarized components of the
particle backscattering coefficient, was calculated and calibrated in the same way as described in
Freudenthaler et al. (2009). Additionally, the LILAS system is capable of profiling the laser-
induced fluorescence of aerosol particles. This is achieved by using a wideband interference filter
with a width of 44 nm, centered at 466 nm, as suggested by Veselovskii et al. (2020). Due to the
strong sunlight background during daytime, the fluorescence observations are limited to nighttime
hours.
The calculation of the fluorescence capacity $G_F$ can be performed using backscattering
coefficients at any laser wavelength. In our study, we specifically used $\beta_{532}$, as it is determined
using rotational Raman scattering and is considered to be the most reliable, thus $G_F = \dfrac{\beta_F}{\beta_{532}}$. To
supplement our measurements, additional information about atmospheric properties was obtained
from radiosonde measurements conducted at Herstmonceux (UK) and Beauvechain (Belgium)
stations, which are located approximately 160 km and 80 km away from the observation site,
respectively. The lidar measurements were primarily conducted vertically. In cases where
observations were made at an angle to the horizon, the corresponding information has been
included in the captions of the figures.

**2.2. Approach for the mixture partition**
The lidar system measures up to nine independent properties of aerosols. However, our
main focus is on separation the backscatters of individual aerosol types with high spatiotemporal
resolution. To calculate parameters related to the extinction coefficient, such as lidar ratio or
extinction Angstrom exponent, it is necessary to average lidar profiles over a substantial
spatiotemporal interval. In this study, as a first step, we use two parameters with high resolution
in both height and temporal domains: the depolarization ratio $\delta_{532}$ and the fluorescence capacity
$G_F$. Moreover, the calculation process partially cancels out the overlap functions, allowing us to
derive $\delta_{532}$ and $G_F$ closer to the ground compared to aerosol extinction. We are considering a
scenario where only three externally mixed aerosol types occur, such as smoke (s), dust (d), and



urban (u). The aerosol and fluorescence backscattering coefficients ($\beta_{532}$ and $\beta_F$) are the sum of
their respective contributions.
$\beta_{532} = \beta_{532}^s + \beta_{532}^d + \beta_{532}^u$           (1)
$\beta_F = \beta_F^s + \beta_F^d + \beta_F^u$           (2)
The fluorescence capacities for each aerosol type are:
$G_F^i = \dfrac{\beta_F^i}{\beta_{532}^i}$           (3)
where $i = s, d, u$. The fractions of $\beta_{532}$ for individual aerosol types are:
$\eta_i = \dfrac{\beta_{532}^i}{\beta_{532}}$    .           (4)
By definition:
$\eta_s + \eta_d + \eta_u = 1.$           (5)
The fluorescence capacity can be expressed as a linear combination of the fluorescence
capacities of each aerosol type, as shown in Eq. 6:
$G_F = \eta_s G_F^s + \eta_d G_F^d + \eta_u G_F^u$           (6)
The particle depolarization ratio is a ratio of the cross- and co-polarized component of the
backscattering coefficient: $\delta_{532} = \dfrac{\beta_{532}^{\perp}}{\beta_{532}^{\parallel}}$ . However, for the mixture analysis, the use of the
depolarization potential $\delta_{532}' = \dfrac{\delta_{532}}{1+\delta_{532}}$ is preferable, because $\delta'$, the same as $G_F$, is a linear
combination of the depolarization potentials of individual particle types ($\delta_{532}'^{s}, \delta_{532}'^{d}, \delta_{532}'^{u}$), as outlined
by Burton et al. (2014).
$\delta_{532}' = \eta_s \delta_{532}'^{s} + \eta_d \delta_{532}'^{d} + \eta_u \delta_{532}'^{u}$           (7)

142       Finally, we have a system of three equations (5-7) from which we can determine the relative

contributions of each aerosol type by finding $\eta_s$, $\eta_d$ and $\eta_u$. In our study, we solve the system (Eq.
5-7) using the least squares method with an additional constraint on the non-negativity of solutions.
To achieve equal weighting of Eq.6 and 7, each equation is scaled by a factor so that the Euclidean
norm of the coefficients $G_F^s, G_F^d, G_F^u$ and $\delta_{532}'^{s}, \delta_{532}'^{d}, \delta_{532}'^{u}$ (considered as a 3-element vectors) become
equal to 1. As mentioned earlier, the particle parameters may vary within predetermined ranges,



even for a specific aerosol type. However, the exact values of $G_F^i$ and $\delta_{532}'$ at a specific height/time
pixel are unknown. To address the uncertainty in $\eta_i$, we employ the Monte Carlo technique,
assuming that $G_F^i$ and $\delta_{532}'$ are random values uniformly distributed within the predetermined
intervals. For each Monte Carlo trial, random values of $G_F^i$ and $\delta_{532}'$ are generated. Instead of
relying on a single solution, we conduct a series of Monte Carlo trials in order to obtain a set of
solutions and calculate the average of this set. The dispersion of these solutions is taken as a
measure of method uncertainty. The number of Monte Carlo trials was set to 100 and further
increase in this number did not significantly impact either the final average or the dispersion of
solutions. In our classification scheme, we include four types of aerosols (smoke, pollen, urban,
dust). Nevertheless, the system of equations (Eq. 5-7) consists of only three equations. Given that
it is highly unlikely to have all four aerosol types coexisting at a single height/time pixel, one of
the four types can be excluded a priori based on a $G_F$-$\delta_{532}$ diagram or other pertinent
considerations. Another option is to exclude one aerosol type at each height/time pixel based on
the lidar data itself, as described below. Such method we will call Automatic Type Selection (ATS)

162        For ATS, we solve the system Eq. 5-7 for the triplets (*S, P, U*), (*S, P, D*), (*S, D, U*), and (*P,*

*D, U*), where *S, D, U, P* denote Smoke, Dust, Urban, Pollen, respectively. To determine which
aerosol types can be excluded, we use the discrepancy for Eq. 6 and 7 as a criterion. Specifically,
we calculate the difference between the input data ($G_F$-$\delta_{532}$) and the corresponding values obtained
by substituting the solution into the right-hand side of Eq. 6 and 7. These two differences are
treated as a 2-element vector, and the Euclidean norm of this vector is taken as the discrepancy.
The aerosol triplet that provides the least discrepancy is chosen for this single Monte Carlo trial
and for the height/time pixel. This procedure is repeated for every Monte Carlo trial, and after
averaging, the spatiotemporal distributions of $\eta_s$, $\eta_p$, $\eta_u$, and $\eta_d$ are evaluated.

**3. Application of partition algorithm to lidar observations**

173        The uncertainty of the partitioning of backscattering coefficients depends on the range of

$G_F$ and $\delta_{532}$ variations in each aerosol type. To establish this range, we analyzed measurement
sessions at the ATOLL for the period of 2020-2023. Our focus was on observation episodes
characterized by stable atmospheric conditions, where only a single aerosol type predominated, at
least within specific height/time intervals. Moreover, we took precautions to ensure that the





relative humidity in the selected intervals remained below 60% to minimize the impact of particle
hygroscopic growth. Based on the obtained results, we summarized the ranges of parameter
variation in Table 1. The depolarization ratios $\delta_{532}$ for smoke and urban particles fall within the
range of 2%-8%, while for dust, this range is 25%-35%. The depolarization ratio of long
transported dust can be lower, but at this stage, we do not consider possible modifications of dust
properties during transportation. We attribute lower values of $\delta_{532}$ to the mixing of dust with
pollutants (urban aerosol in our model). The fluorescence capacity of smoke in the upper
troposphere can be as high as $10 \times 10^{-4}$ (Veselovskii et al., 2023), but below 8 km, it mainly falls
within the range of $(2.5-4.5) \times 10^{-4}$. For dust and urban particles, the values of fluorescence
capacities are within the intervals of $(0.05-0.45) \times 10^{-4}$ and $(0.2-0.8) \times 10^{-4}$, respectively.
Determining the ranges of $\delta_{532}$ and $G_F$ for pollen is particularly challenging because, in the North
of France, pollen is commonly mixed with other aerosol types. Moreover, the depolarization of
pollen particles varies significantly from one type to another (Cao et al., 2010). In the Lille area,
one dominant taxon is birch (Veselovskii et al., 2021) with a depolarization ratio of $\delta_{532}$ at around
30% (Cholleton et al., 2022). In our analysis, the depolarization ratio is set within the 30%-40%
interval. The variation range of $G_F$ is estimated from our measurements to be within $(1.0-2.5) \times 10^{-4}$
$^4$.
Table 1. Variation ranges of fluorescence capacity and the particle depolarization ratio for different
types of aerosols.

| Type | $G_F$, $10^{-4}$ | $\delta_{532}$, % |
|------|------|------|
| Smoke | 2.5÷4.5 | 2.0÷8 |
| Pollen | 1÷2.5 | 30÷40 |
| Urban | 0.2÷0.8 | 2.0÷8 |
| Dust | 0.05÷0.45 | 25÷35 |


Below, we present two examples of applying the described approach to measurements performed
on March 27-28, 2022, and October 1-2, 2023.





**March 27-28, 2022**


The spatiotemporal distributions of the aerosol backscattering coefficient $\beta_{532}$, the particle
depolarization ratio $\delta_{532}$, and the fluorescence capacity $G_F$ on March 27-28, 2022, are shown in
Fig.1. Relative humidity decreased with height, ranging from 70% at 600 m to 55% at 1800 m.
Aerosols were primarily found below 2500 m, with several distinguishable particle types identified.
The particle depolarization ratio increased to 30% at 2000 m during the 20:00-22:00 UTC period,
indicating the presence of dust. Additionally, high values of the fluorescence capacity (up to
$2.5\times10^{-4}$) for the 00:00-05:00 UTC period suggest the presence of smoke.
Fig.2a presents the $G_F$-$\delta_{532}$ diagram for these measurements. The red boxes represent the
parameter ranges used for aerosol classification, which are slightly broader than those outlined in
Table 1 to account for mixtures where one type is predominant. Dust, smoke, and urban particles
can be distinguished in the clusters of points on the diagram, with intervals indicating mixed
particle types. Although March is typically a pollen season in Lille, pollen particles did not
significantly contribute to the observed episode. Utilizing this classification scheme, we assess the
spatiotemporal distribution of aerosol types in Fig.2b, following the methodology outlined in
Veselovskii et al. (2022). Regions predominated by dust, smoke, and urban particles are clearly
identified. A small amount of pollen is observed towards the end of the session at approximately
700 m height. The grey color in Fig.2b represents aerosol mixtures where the particle type cannot
be definitively identified. The aerosol classification presented in Fig. 2b finds support in the results
of the HYSPLIT Backward Trajectory Analysis (Stein et al., 2015) depicted in Figure 3.
Specifically, the air masses below 1000 m height were transported over the Belgium, and the
presence of urban aerosol is expected. Conversely, the air masses above 1500 m were transported
over regions with extensive forest fires in Greece and near Spain, suggesting a potential mixture
of smoke and dust.
By applying the partition technique described in Sect.2.2, we can determine the contribution
of each particle type to the total backscattering coefficient $\beta_{532}$. The spatiotemporal distributions of
$\eta_s$, $\eta_u$, and $\eta_d$ in Fig.4 were assessed assuming that pollen contribution can be neglected. The
algorithm operates smoothly, showing distributions without any unrealistic high-frequency
oscillations. By observing the distributions, it can be concluded that the smoke plume actually
contains a significant amount of urban aerosol, while the dust plume does not show the presence
of other particle types.



The distributions in Fig.4 represent the mean values of $\eta_s$, $\eta_u$, and $\eta_d$. To understand the
uncertainty caused by potential variations in particle characteristics, Fig.5 displays the vertical
profiles of $\eta_s$, $\eta_u$, and $\eta_d$ for the period between 21:00-22:00 UTC, along with the corresponding
standard deviations. Urban particles are predominant below 1000 m with a deviation from the
mean value of roughly 5%. Above 1500 m, $\eta_u$ decreases to 0.05 and the uncertainty increases to
100%. Conversely, dust can be disregarded below 1000 m, but becomes predominant above 1000
m. Smoke contribution during the considered time period is low and only becomes noticeable
($\eta_s$~0.15) in the 1250-1500 m range. As mentioned earlier, the results in Fig. 4 were obtained
without considering pollen. To assess the potential impact of pollen on the results, the partition
was carried out for four aerosol types using the ATS approach, as described in Section 2.2. The
corresponding profiles of $\eta_{s,4}$, $\eta_{u,4}$, and $\eta_{d,4}$, are depicted in Fig.5 with magenta lines. Notably, the
profiles obtained for three and four aerosol types are similar. Pollen does have some effect on
smoke contribution ($\eta_s$ decreased from 0.14 to 1.0), but its influence on dust and urban particle
contribution is negligible.

**October 1-2, 2023**

Observations at ATOLL in 2023 were notable for frequent intensive smoke events. North
American wildfire smoke, transported over the Atlantic, was observed from mid-May until
October. In some autumn episodes, smoke descended from the troposphere to ground level. One
such episode is shown in Fig.6, which presents the spatiotemporal distributions of $\beta_{532}$, $\delta_{532}$, and
$G_F$ during the night of October 1-2, 2023. During this period, the relative humidity decreased with
height, from 50% at 500 m to 30% at 3500 m. Strong aerosol layers were observed up to 5 km in
height, and the depolarization ratio $\delta_{532}$ exceeded 25% above 2000 m, indicating the predominance
of dust. However, below 1000 m, a low depolarization ratio ($\delta_{532} < 8\%$) was accompanied by a
high fluorescence capacity of particles (up to $3.0\times10^{-4}$), identifying them as smoke. The $G_F$-$\delta_{532}$
diagram in Fig.7a highlights the pixels attributed to dust, smoke, and urban particles. There are
also intervals where these types were mixed. These regions with mixed aerosols are represented
by the grey color in the distribution of particle types in Fig.7b. The results of aerosol classification
agree with HYSPLIT backward trajectories analysis. Fig.8 shows the five-days back trajectories
over Lille on October 2, 2023, at 00:00 UTC. The air masses over the Atlantic, containing North
American smoke, descend from 5000 m to the ground, leading to the predominance of smoke over



Lille at 500 m. The air masses at 1500 m are transported over the continent and may contain
pollutants, whereas the air masses at 2700 m arrive from Africa and are loaded with dust. Fig. 9
depicts the spatiotemporal distributions of $\eta_s$, $\eta_u$, $\eta_d$, derived in assumption that only three aerosol
types occur. Urban aerosol is localized primarily between the smoke and dust layers. Vertical
profiles of $\eta_s$, $\eta_u$, $\eta_d$ for the 22:00-23:00 UTC period are presented in Fig.10. Smoke predominates
below 1000 m, with a smoke contribution ($\eta_s$=0.7 at 750 m) evaluated with an uncertainty of about
20%. The contribution of urban particles within the smoke layer (at 750 m) is $\eta_u$=0.3, with a
corresponding uncertainty of approximately 30%. Dust predominates above 2000 m ($\eta_d$=0.8), and
the uncertainty of $\eta_d$ estimation is below 15%. Although the existence of pollen in October is quite
improbable, for testing purposes, we performed an inversion for four aerosol types using the ATS
method (magenta lines in Fig.10). The impact of including pollen is most pronounced for dust at
1750 m, where $\eta_d$ is about 25% decreased. However, the values obtained still fall within the
estimated range of uncertainty. From the examples considered, we conclude that the contributions
of three aerosol components to the backscattering coefficient can be determined through joint
fluorescence and polarization measurements. The volume density, $V_i$, of i-th aerosol component
can be estimated from the backscattering coefficient using the corresponding lidar ratio, $S_{532}^i$, and
the extinction-to-volume conversion factors $C_V^i$ (Mamouri and Ansmann, 2017; Ansmann et al.,
2019, 2021; He et al., 2023). Thus, for the i-th aerosol component:
$$V_i = \beta_{532} \times \eta_i \times S_{532}^i \times C_V^i \qquad (8)$$
The values of the conversion factors at 532 nm, derived from AERONET observations, along with
some reported lidar ratios, are summarized in Table 2. Therefore, the presented information allows
us to quantify the composition of the aerosol mixture.





Table 2. Lidar ratios ($S_{532}^i$) and extinction-to-volume conversion factors ($C_V^i$) for different types
of aerosol.

| Type | Lidar ratio $S_{532}^i$, sr | $C_V^i$, µm$^3$cm$^{-3}$/Mm$^{-1}$ |
|---|---|---|
| Urban | 53-70 [5] | 0.3-0.41 [1] |
| Smoke (North American, aged) | 55-73 [5] | 0.13 [3] |
| Dust (North Africa) | 40-50 [3] | 0.61-0.64 [1] <br> 0.67-0.73 [2] <br> 0.64-0.67 [4] |

[1] Mamouri and Ansmann, 2017; [2] Ansmann et al., 2019; [3] Ansmann et al., 2021; [4] He et al., 2023; [5] Burton et
al., 2013

## 291    4. Heatwave over Lille in July 2022.

The heatwave in France in July 2022 was attributed to a high-pressure system known as the
Azores High, which usually sits off Spain and pushed farther north, resulting in elevated
temperatures and multiple fires. The Sun photometer and lidar observations at ATOLL consistently
recorded an increase in aerosol content over Lille in the middle of July 2022. Fig.11 displays the
aerosol optical depth (AOD) at 500 nm and the Angstrom exponent for 380/500 nm wavelengths
provided by AERONET. Lidar observations were performed from July 16 to July 23, as shown in
the frame in Fig.10. Within this interval, the optical depth increased, reaching its peak on July 18.
The Angstrom exponent decreased, indicating the presence of dust. Fig.11 shows the column-
integrated particle volume, provided by AERONET, presented separately for the fine and coarse
mode particles. After July 16, the volume of the coarse mode increased approximately fourfold,
while the fine mode did not show significant changes, further supporting the presence of dust
particles. Unfortunately, volume retrievals are not available after July 20 due to the presence of
clouds. The methodology outlined in Sect. 2.2 was used to analyze the composition of aerosols
during the heatwave.
In Fig.13, we can see the spatiotemporal distributions of $\beta_{532}$, $\delta_{532}$ and $G_F$ for four
measurement sessions between July 16 and July 23, 2022. On July 16-17, after midnight, a dust
layer with $\delta_{532}$ exceeding 20% appeared at a height of 5 km. The following night (July 17-18), the
lower boundary of the dust layer descended to 2 km. By the night of July 18-19, we observed



strong aerosol backscattering (above 1.0 Mm$^{-1}$sr$^{-1}$) from the ground up to a height of 5 km. Dust
was primarily found within two height ranges: 0.75-2.0 km and 3.0-5.0 km, where the particle
depolarization ratio $\delta_{532}$ exceeded 20%. The aerosol between these dust layers showed high
fluorescence capacity (above $2.0\times10^{-4}$), indicating the presence of smoke. Unfortunately, we could
not make long-term lidar observations from July 19-21 due to cloud cover. However, by the night
of July 22-23, we observed localized aerosols below 3 km. The values of $\delta_{532}$ and $G_F$ were below
10% and $1.0\times10^{-4}$, respectively, which is typical for urban particles. The relative humidity during
the measurements for July 16-19 was below 60 % within the height range being considered. On
the night of July 22-23, the relative humidity was higher, reaching up to 80%. In Fig.14, we provide
the $G_F$-$\delta_{532}$ diagrams for the measurements shown in Fig.13. On the night of July 16-17, the
clusters corresponding to dust and smoke/urban particles are distinct. However, for July 17-19,
dust was mixed with smoke and urban particles, resulting in a characteristic pattern on the $G_F$-$\delta_{532}$
diagram (Veselovskii et al., 2022). By the night of July 22-23, only one cluster, corresponding to
urban aerosol, was observed. The distributions of particle types in Fig.14 for the period of July 16-
19 contain extended gray regions where different types of particles are mixed and cannot be
identified. In Fig.15, we can see the partition technique used to evaluate the contributions of dust,
smoke, and urban aerosol to $\beta_{532}$. From this analysis, we can conclude that on the night of July 16-
17, the aerosol below 2.5 km was a mixture of smoke and urban particles, and the elevated dust
layer (00:00-03:00 UTC) contained a significant amount of urban particles ($\eta_u$ is up to 0.4). On
July 18-19, the aerosol between the two dust layers, within the height range of 2-3 km, was also a
mixture of smoke and urban particles.
The aerosol classification based on fluorescence and depolarization measurements is
supported by the analysis of backward trajectories. Fig.16 shows the five-day backward
trajectories for four measurement sessions from Figure 15 at altitudes of 1500 m, 3000 m, and
4500 m. On July 16-17, the dust layer above 4000 m originates from North Africa, while smoke
at 3000 m is likely transported from North America. The air masses at 3000 m on July 17-18 are
transported from Africa over regions of wildfires in Spain, indicating a mixture of dust and smoke.
Smoke at 3000 m on July 18-19 again originates from wildfires in Spain, while the source of the
dust layers at 1500 m and 4000 m is in Africa. Finally, on July 22-23, the air masses arrive from
the West outside dust and smoke sources, and aerosol in Fig. 15 within the 1000-3000 m range is
identified as urban.





As mentioned, the volume density of each component can be estimated using Eq. 8. Fig.17
presents the vertical profiles of volume density for smoke, urban, and dust particles for four
measurement sessions from Fig.15. In the calculations, we used the mean values of $\eta_s$, $\eta_u$, $\eta_d$, as
well as the mean values of the lidar ratios and fluorescence capacity from Table 2. The lidar ratios
for smoke, urban, and dust are 64 sr, 61 sr, and 45 sr, respectively, and the fluorescence capacity
values are $0.13\times10^{-4}$, $0.35\times10^{-4}$, and $0.7\times10^{-4}$, respectively. The main contributors to the volume
are urban and dust particles, with smoke contributing noticeably only on July 18 and 19, but with
a volume density still below 5 $\mu m^3 cm^{-3}$. To assess the validity of our volume estimations, we
compared our results with AERONET retrievals. For this comparison, the volume profiles of each
component from Fig.17 were extrapolated to the ground, and the total column-integrated volume
was calculated. The results are depicted in Fig.12 by stars, with an additional measurement on July
19 (22:00-23:00) included. It is evident that the results provided by AERONET are in reasonable
agreement with the results provided by the lidar.

**Conclusion**
In conclusion, this study introduces an approach to partition aerosol mixtures in terms of
backscattering coefficients, based on fluorescence and polarization lidar measurements.
Specifically, we used the particle depolarization ratio at 532 nm and the fluorescence capacity,
allowing for the partitioning of a three-component aerosol mixture at every height/time pixel. The
robustness of this approach is demonstrated through testing with Mie-Raman-fluorescence lidar
observations at the ATOLL instrumental site, providing valuable insights into the composition and
dynamics of atmospheric aerosols. One notable advantage of the proposed approach is its
applicability even in conditions of low aerosol content or for aerosol layers in the upper
troposphere, where deriving profiles of extinction coefficients might be challenging. Additionally,
backscattering coefficients of aerosol components can be converted to particle volume densities
using corresponding lidar ratios along with extinction-to-volume conversion factors. While this
conversion provides a rough volume estimation, considering the variability of the lidar ratios and
the conversion factors within a given aerosol type, a comparison of lidar-derived particle volume
during the heatwave over Lille in July 2022 demonstrates promising agreement with AERONET
retrievals. At this stage, we have simplified our classification scheme by incorporating four aerosol
types: smoke, dust, pollen, and urban particles. It is important to note that the use of fluorescence



is an efficient way to distinguish between urban and smoke particles, which is a challenge for other
methods that do not utilize fluorescence. However, we recognize the need to expand our approach
to include additional aerosol types, particularly those with strong absorption such as polluted urban
aerosol. This expansion will involve incorporating additional particle parameters, like lidar ratios,
and is planned for our future research. It is crucial to acknowledge that the particle hygroscopic
growth complicates the use of fluorescence capacity, resulting in increased uncertainty. To address
this, we aim to utilize the additional independent information about aerosol type provided by the
fluorescence spectrum. Importantly, the fluorescence spectrum is not affected by relative humidity.
In our future research, we plan to further enhance the fluorescence capabilities by increasing the
number of fluorescence channels in the lidar.

*Data availability*. Lidar measurements are available upon request
(philippe.goloub@univ-lille.fr).

*Author contributions*. IV processed the data and wrote the paper. BB prepared the program for
aerosol mixture partitioning.  QH performed meteorological analysis. TP, GD and WB performed
lidar measurements in Lille. PG supervised the project and helped with paper preparation. MK and
NK participated in algorithms development and data analysis.
390    .

*Competing interests*. The authors declare that they have no conflict of interests.

### Acknowledgement

We acknowledge funding from the CaPPA project funded by the ANR through the PIA under
contract ANR-11-LABX-0005-01, the "Hauts de France" Regional Council (project ECRIN) and
the European Regional Development Fund (FEDER). ESA/QA4EO program is greatly
acknowledged for supporting the observation activity at LOA.  The work from Q. Hu was
supported by Agence *Nationale* de Recherche ANR (*ANR-21-ESRE-0013) through the*
OBS4CLIM project. Development of algorithms was supported by Russian Science Foundation
(project 21-17-00114).

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

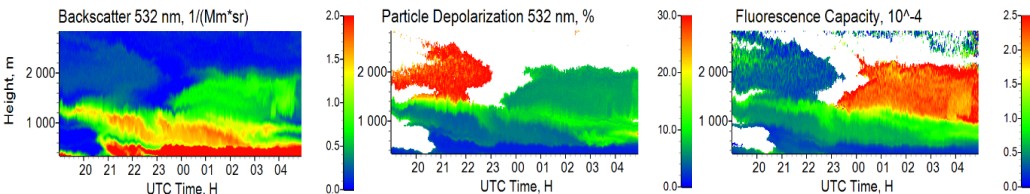


Fig.1. Spatiotemporal distributions of the backscattering coefficient at 532 nm, particle
depolarization ratio at 532 nm and fluorescence capacity during the night of March 27-28, 2022.
The depolarization ratio and fluorescence capacity are calculated only for the values $\beta_{532} > 0.1$ Mm$^-$
$^1$sr$^{-1}$. The measurements were taken at an angle of $45^0$ to the horizon.




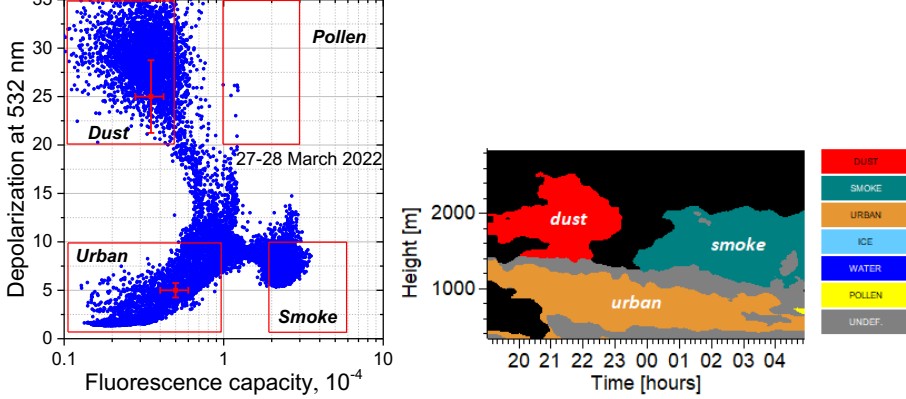


Fig.2. (a) The $\delta_{532}$-$G_F$ diagram for observations in the height range of 3500 m–2800 m and (b) the
spatiotemporal distribution of aerosol types during the night of March 27–28, 2022.





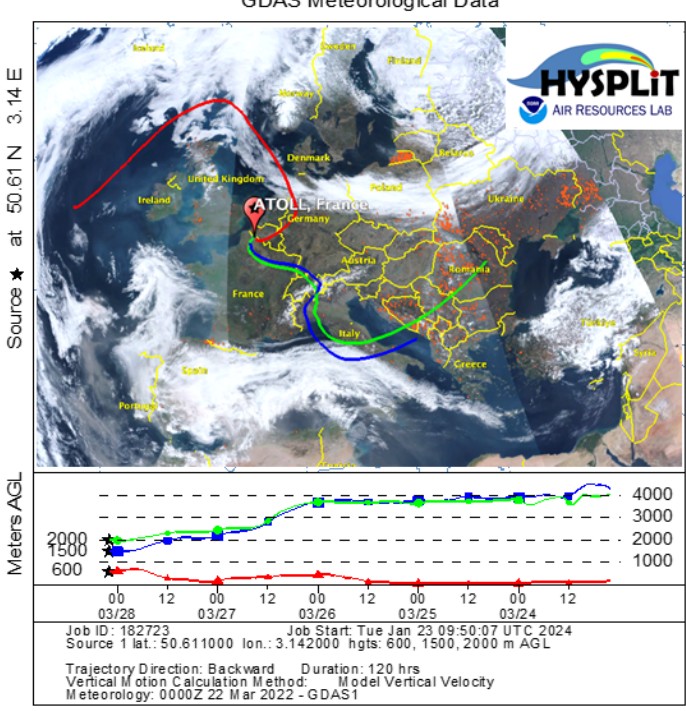


Fig.3. The HYSPLIT five-day backward trajectories for the air mass over Lille at altitudes 600 m,

1500 m, and 2000 m on March 28, 2022 at 02:00 UTC. Red dots depict the regions of forest fires.




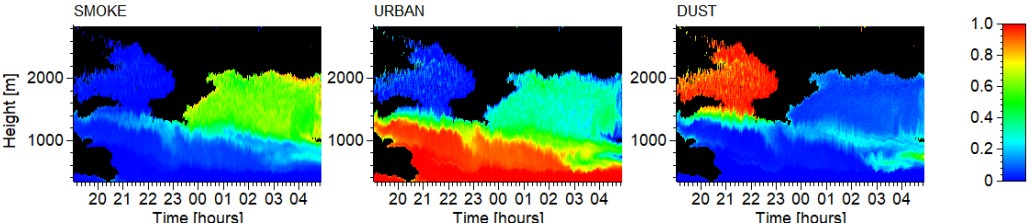


Fig.4. Relative contributions of smoke ($\eta_s$), urban ($\eta_u$), and dust ($\eta_d$) particles to the backscattering

coefficient $\beta_{532}$ during the night of March 27–28, 2022.





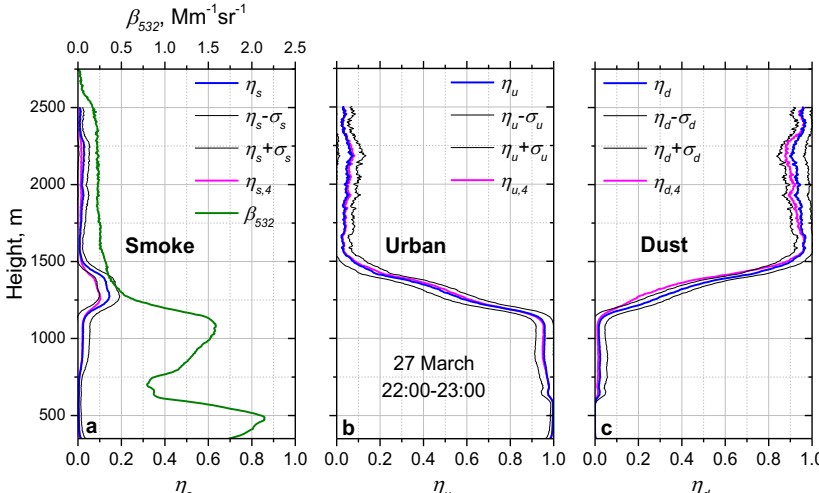


Fig.5. Vertical profiles of the relative contributions of smoke ($\eta_s$), urban ($\eta_u$), and dust ($\eta_d$) particles to the backscattering coefficient $\beta_{532}$ on March 27, 2022. These profiles are derived under the assumption that only three aerosol types occur. The black lines depict the deviation of solutions from the mean value ($\eta_i \pm \sigma_i$). Magenta lines show the relative contributions of smoke, urban and dust particles ($\eta_{s,4}$, $\eta_{u,4}$, $\eta_{d,4}$) when four aerosol types (including pollen) are considered.

550





551

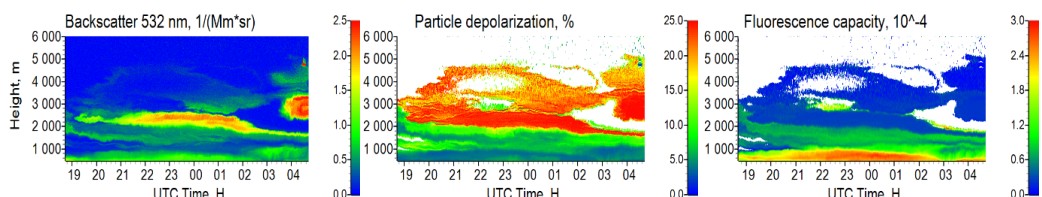

552

Fig.6. Spatiotemporal distributions of the backscattering coefficient at 532 nm, particle depolarization ratio at 532 nm and fluorescence capacity during the night of October 1-2, 2023. The depolarization ratio and fluorescence capacity are calculated only for values of $\beta_{532}$>0.1 Mm$^{-1}$sr$^{-1}$.


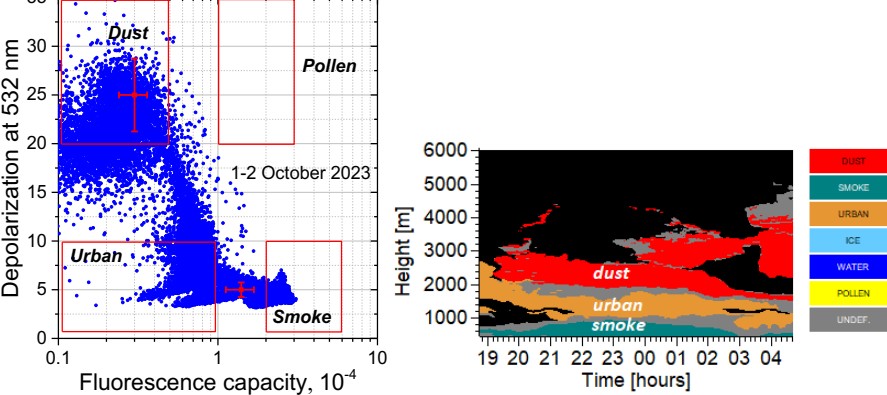


Fig.7. (a) The $\delta_{532}$-$G_F$ diagram and (b) the spatiotemporal distribution of aerosol types during the night of October 1-2, 2023.



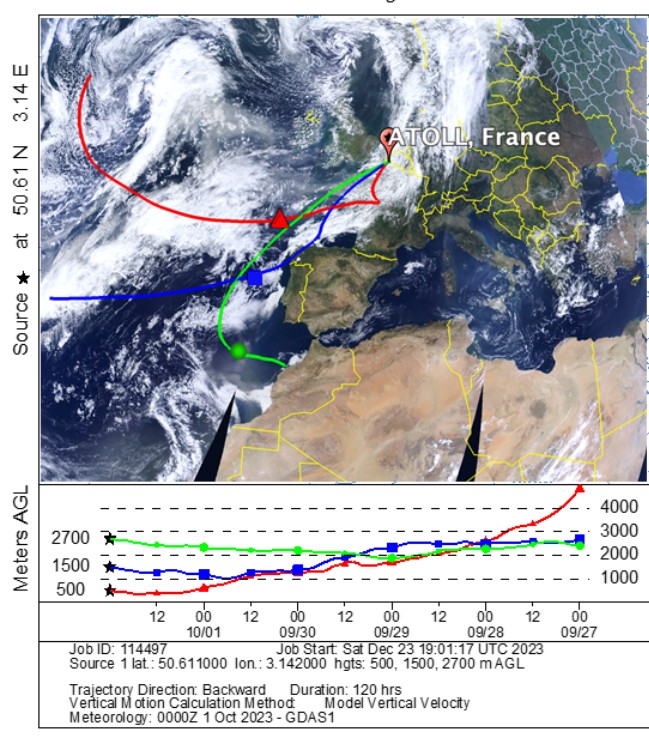


Fig.8. The HYSPLIT five-day backward trajectories for the air mass over Lille at altitudes 500 m,

1500 m, and 2700 m on October 2, 2023 at 00:00 UTC.





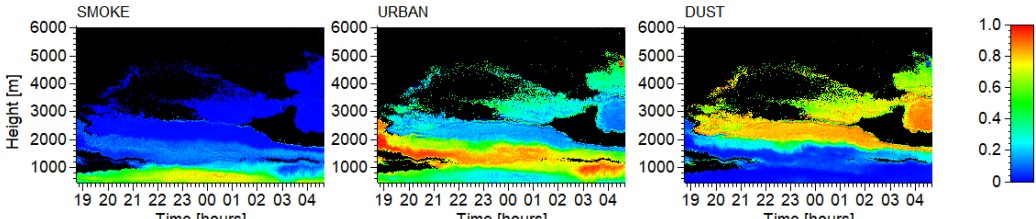


Fig.9. The relative contributions of smoke ($\eta_s$), urban ($\eta_u$), and dust ($\eta_d$) particles to the backscattering coefficient $\beta_{532}$ during the night of October 1-2, 2023.





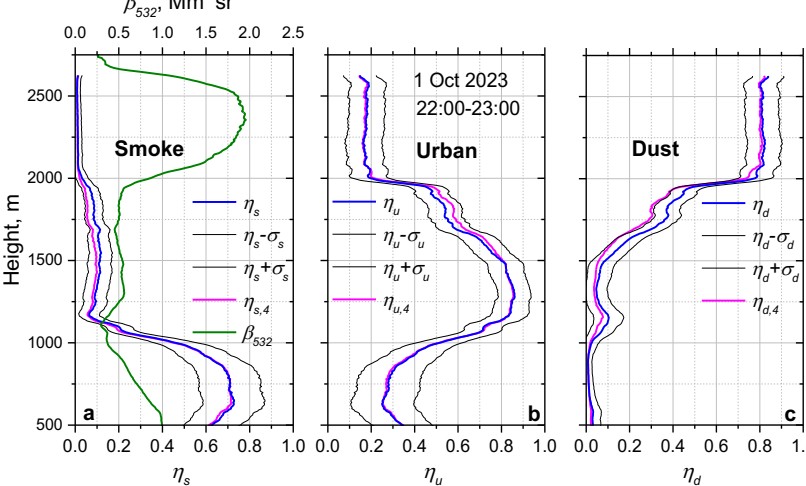


Fig.10. Vertical profiles of the relative contributions of smoke ($\eta_s$), urban ($\eta_u$), and dust ($\eta_d$) particles to the backscattering coefficient $\beta_{532}$ on October 1, 2023. The profiles are derived under the assumption that only three aerosol types occur. The black lines depict the deviation of solutions from the mean value ($\eta_i \pm \sigma_i$). The magenta lines show the relative contributions of smoke, dust and urban particles ($\eta_{s,4}$, $\eta_{u,4}$, $\eta_{d,4}$) when four aerosol types (including pollen) are considered.





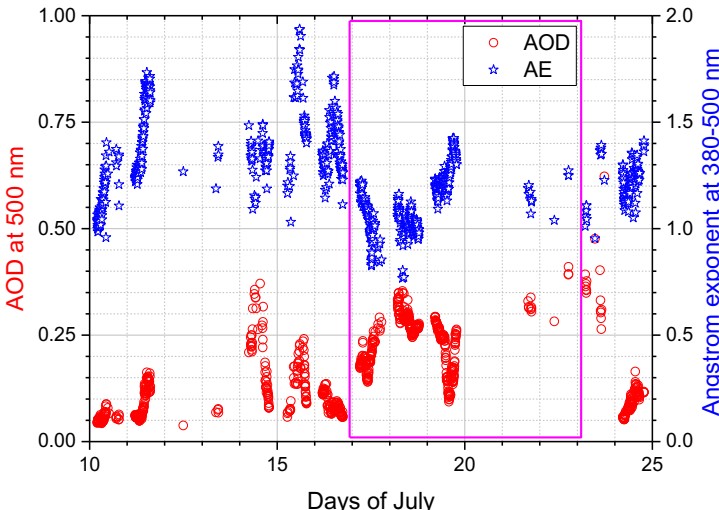


Fig.11. The aerosol optical depth (AOD) at 500 nm and the Angstrom exponent (AE) provided by

AERONET over Lille in July 2022. Magenta box depicts the time period during which lidar

observations in this study were analyzed.

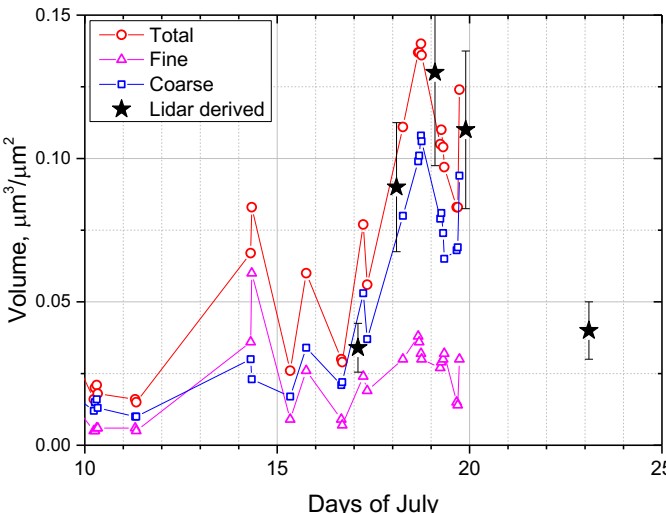


Fig.12. Column-integrated aerosol volume (circles) in July 2022 provided by AERONET. The

triangles and squares represent the volumes of the fine and coarse modes, respectively. Black stars

depict the particle volume derived from lidar observations.






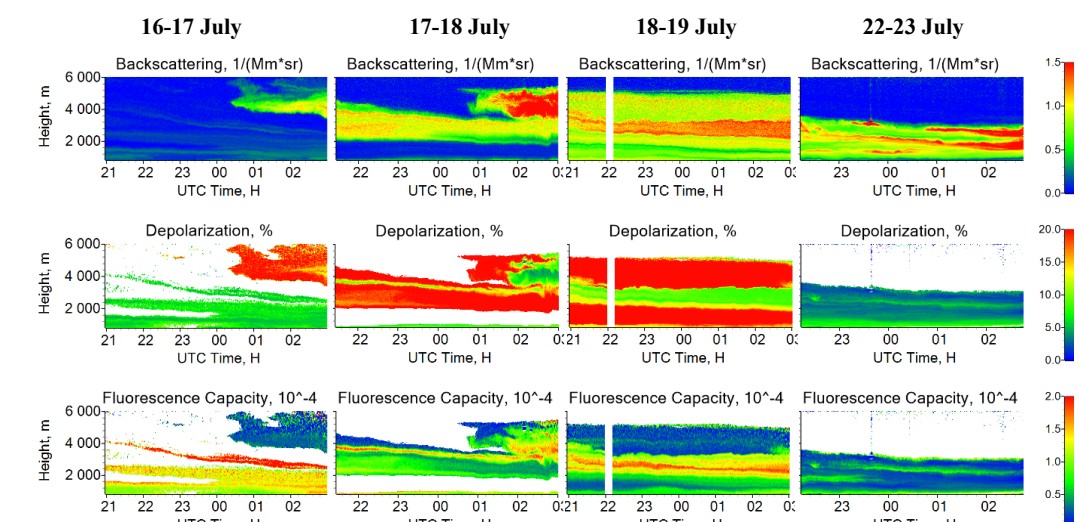

Fig.13. Spatiotemporal distributions of the backscattering coefficient $\beta_{532}$, the particle depolarization ratio $\delta_{532}$, and the fluorescence capacity $G_F$ for the nights of July 16-17, 17-18, 18-19 and 22-23, 2022. The depolarization ratio and fluorescence capacity are calculated only for the values $\beta_{532} > 0.1$ Mm$^{-1}$sr$^{-1}$.








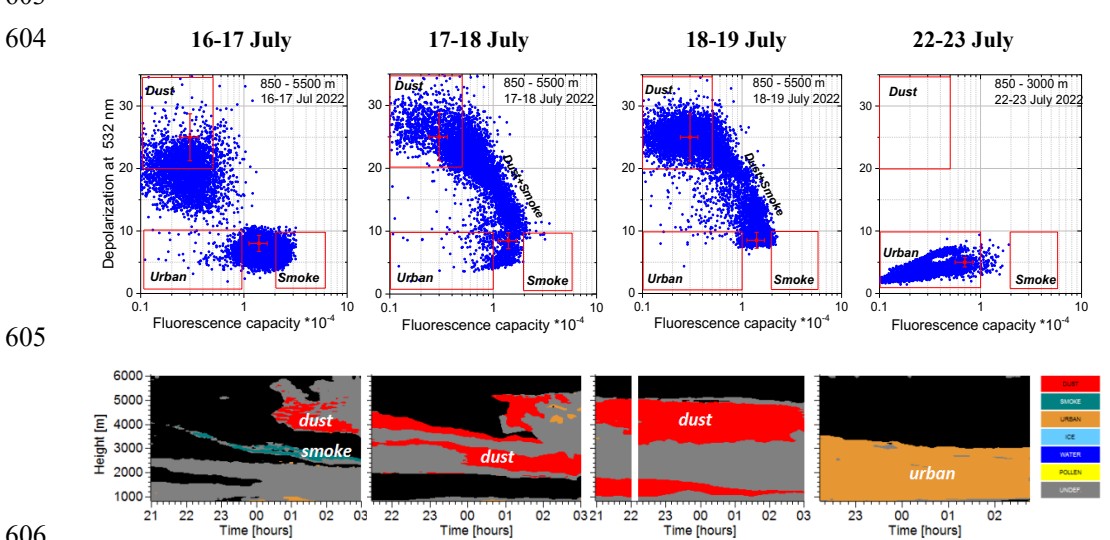



Fig.14. The $\delta_{532}$-$G_F$ diagram (upper row) and the spatiotemporal distribution of aerosol types
(bottom row) for the measurements for the nights of July 16-17, 17-18, 18-19 and 22-23, 2022.
The grey coloring represents an undefined aerosol type.




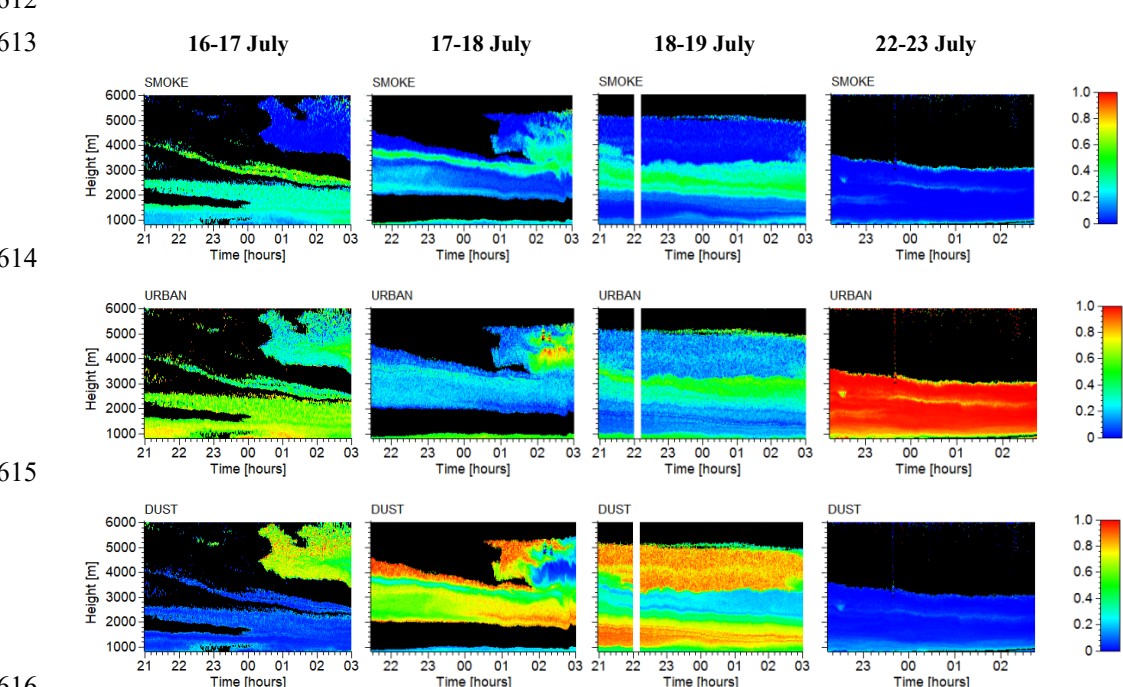




Fig.15. The relative contributions of smoke, urban and dust particles to the backscattering
coefficient at 532 nm for the nights of July 16-17, 17-18, 18-19 and 22-23, 2022.







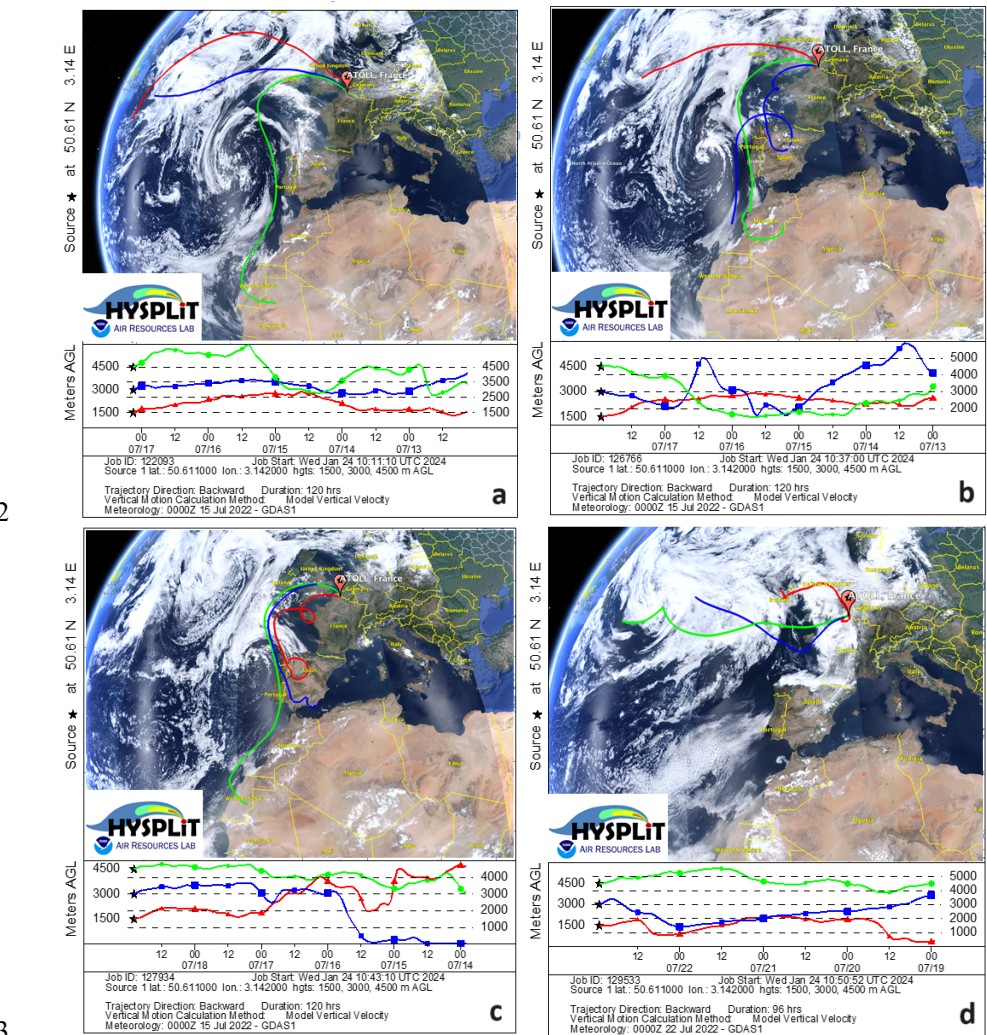


Fig.16. The HYSPLIT five-day backward trajectories for the air mass over Lille at altitudes 1500
m, 3000 m, and 4500 m on (a) July 17, 2022 at 03:00 UTC; (b) July 17, 2022 at 23:00 UTC; (c)
July 18, 2022 at 22:00 UTC; (d) July 22, 2022 at 22:00 UTC. Red dots depict the regions of forest
fires.






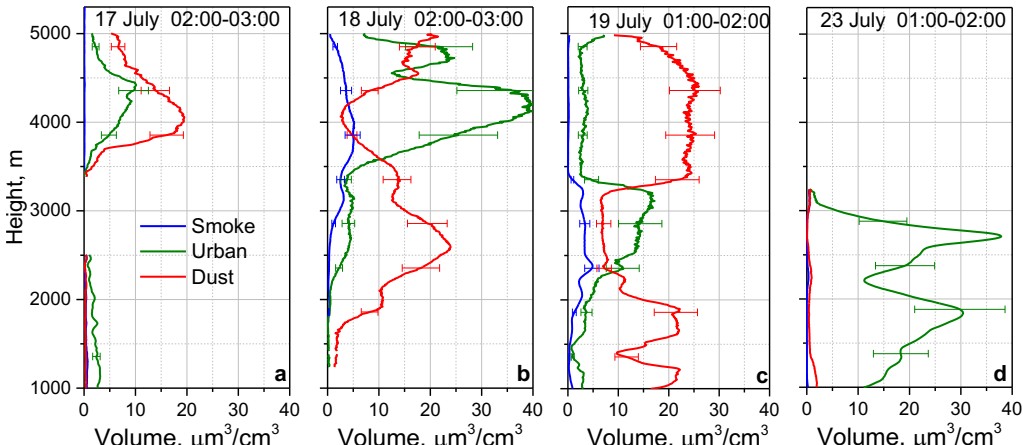


Fig.17. Vertical profiles of the volume density of smoke, dust and urban particles derived from $\eta_s$,
$\eta_u$, and $\eta_d$ presented in Fig.13, using the mean values of the lidar ratios and the conversion factors
from Table 2.