# Peer review of "Retrieval and analysis of the composition of an aerosol mixture through Mie-Raman Fluorescence lidar observations."

_Atmospheric Measurement Techniques, 2024_

## Referee Comment (RC2)

The work is devoted to the problem of recognizing and identifying the type of aerosol in lidar measurements. It is known that aerosols in the atmosphere arise from numerous sources and undergo intense mixing during atmospheric transport. Typically, the aerosol environment at a particular geographic location is a mixture of different types of particles. In this regard, the task of recognizing aerosol particles by their origin is extremely important. The article under discussion presents an original approach to recognizing the components of aerosol mixtures based on the analysis of polarization and spectroscopic properties of backscattering coefficients. The proposed method uses data obtained with the Mi-Raman fluorescent lidar at the University of Lille, France. This lidar is well known from the authors' past publications and, with its advanced measurement capabilities, provides rich information on aerosols. The authors propose to take the aerosol backscattering coefficient, the particle depolarization coefficient, and the fluorescent ability as the main input parameters to identify the type of aerosol. Fluorescence ability is defined as the ratio of the fluorescence backscatter coefficient to the backscatter coefficient of particles at the un shifted scattering line. In this case, it is possible to write a system of three lidar equations, the solution of which allows one to determine the composition of the three-component aerosol mixture. The work shows that the method allows one to reliably separate the contributions of such types of aerosols as forest fire smoke, urban aerosol, and dust into the total backscattering coefficient at a wavelength of 532 nm. To achieve this, much work has been done on statistical analysis of the scattering properties of each of these types of aerosols. Obviously, due to the variability of the scattering properties of aerosols, only some statistical relationship is possible between the type of aerosol and the combination of its scattering properties. Understanding this well, the authors use the Monte Carlo method, provided that the scattering properties of aerosols are uniformly distributed within predetermined intervals. This approach provides a statistical sample of possible solutions, allowing the average and dispersion to be calculated over the entire set of solutions. In this case, the dispersion serves as an indicator of the uncertainty of the method. This approach was tested using 28 observations carried out at the ATOLL Observatory, Laboratoire d'Optique Atmosphérique, University of Lille, France.

The presented publication is undoubtedly an important step towards improving lidar methods for their more complete use for studying the dynamics of aerosol layers, atmospheric circulation processes, global transport phenomena, etc. The proposed approach can significantly expand the capabilities of lidar systems and increase the efficiency of lidar networks.

It should be noted that the proposed method has some limitations. There is no doubt that the method works well for dry aerosol and it is quite obvious that it will stop working in case of

wetting of aerosol particles. Fortunately, the authors, being experienced experimenters, understand this and discard measurements made in conditions of relative humidity greater than 60%. However, it is not very clear from the publication how they control the relative humidity along the sounding path. This suggests some recommendation to the authors to supplement their lidar, for example, with Raman channels for measuring temperature and humidity. This modernization is quite possible. In this case, all questions about the applicability of the method would be automatically resolved.

According to formal characteristics, the article fully complies with the requirements of the AMT, since it discusses scientific issues relevant to the topic of AMT. The publication presents new concepts, ideas, tools, and data. Substantial conclusions are drawn and the scientific methods and assumptions are reasonable and clearly stated. The findings are sufficient to support interpretations and conclusions. Descriptions of experiments and calculations are presented quite fully and accurately, and can be reproduced by fellow scientists.  The authors pay due attention to the achievements of colleagues in the field under discussion and clearly indicate their original contributions. The title of the article reflects the content. The abstract contains a brief and complete summary of the content of the article. The overall composition of the article is well structured and understandable. The language of presentation is quite free and precise. Mathematical formulas, symbols, abbreviations, and units of measurement are defined correctly and used as intended. There is no need to shorten, combine or exclude any parts of the article (text, formulas, figures, tables). The number and quality of links correspond to the content of the text. The quantity and quality of additional material meets the requirements. To summarize, it should be noted that the publication under discussion makes the most pleasant impression, is a significant contribution to the development of remote methods for monitoring atmospheric parameters, and can be published without modifications.

---

## Author Comment (AC1)

**Response to Referee 1**

First of all, we would like to thank the Referee for careful reading the manuscript and for numerous useful comments. We followed his recommendations in the process of manuscript revision.

*The paper is well written and appropriate for AMT. It is an excellent contribution the lidar literature.*
*I have only minor points. Since, many publications have been written and published by the Lille lidar team on the application of polarization/fluorescence lidar in atmospheric aerosol research during the last few years, the authors should make very clear (in the Introduction) what is the step forward here (not covered by the foregoing papers). Only by 1:1 comparison with the paper of Veselovskii et al. (AMT, 2022), I saw the difference. The layout of many figures in the new publication (2024) has not changed compared to the 2022 paper. The motivation is given in the Introduction, but not well enough. A more contrasting wording would be helpful. Maybe, start to review the 2022 paper and then state what was missing so far, and this gap is now filled with the 2024 publication.*

We agree with this comment and in revised manuscript the Introduction was modified, to make clear the novelty of this manuscript comparing to our publication in 2022.

*Individual remarks:*

*Section 2.2 is new, and that must be better highlighted in the Introduction.*
Done

*After line 142 it becomes quite complicated (without a nice flow chart of all the steps…). One could even start to explain (step by step) the respective three-aerosol-component separation approach before one continues with the discussion of the methodology in case of the four-component system.*

We tried to skip the mathematical details of the system solving and in the revised manuscript, we tried to make it easier for reader.

*Table 1: The numbers now differ a bit from the ones in Table 1 in Veselovskii et al. (2022). Should be explained!*

Yes, the ranges of parameters variation changed a little, comparing to our 2022 paper, because numerous measurements were performed since that publication, providing more information for establishing these ranges. Also, we limited the ranges by the values which we normally observed in the low and middle troposphere. Comment is inserted in the manuscript.

*Smoke depol values from 2.0-8.0! Does that cover the full range of values. Baars et al. (2019) or Ohneiser et al (2020) show values up to 20% at 532 nm.*

High depolarization ratios of smoke at 532 nm are usually observed in the upper troposphere, while in this manuscript we analyze measurements in low in middle troposphere. For the same reason we limited the maximal value of the fluorescence capacity of smoke at 4.5*10-4, though higher values were observed in upper troposphere.

*Depol values of 2-8% in the case of urban aerosol! Does this range of values (up to 8%) include road dust? Why should there be a depol ratio of >5% in the case of a sulfate-aerosol dominating aerosol?*

The depolarization of urban particles up to 8% we regularly observe over Lille. Contribution of road dust and soil is possible, but at present stage we are not ready to discriminate it.

*Page 8, lines 209-210: Please provide reference to Veselovskii et al. (2022).*
Added

*Page 8, line 223: Spain? I do not see that! You mean: Italy?*
Corrected.

*Page 8, line 244: eta-S = 0.1 and not 1.0*
Corrected

*Page 9, line 250 ... from the free troposphere ... By the way, the 1 October 2023 smoke event was a UNIQUE event. It is almost impossible to find North American smoke so close to the ground. I hope there will be another paper on this UNIQUE observation.*

Yes, we have plans for such paper

*Page 10, line 281-284: I would step forward to mass concentration! Particle densities are 1.15 g/cm3 (smoke), 2.6 g/cm3 (mineral dust), and 1.5g/cm3 (sulfate aerosol). These numbers are given in the referenced papers.*
Done

*Table 2: regarding aged smoke, I would cite own papers as well (Hu et al., .....)*
Added

*Section 4: Why is that an extra section and not simply a subsection of section 3? Please provide a small introduction why you present and discuss this episode separately.*

We agree with reviewer. Section 4 now is subsection of section 3.

*Page 12, line 340: When was the heat wave over? Should be mentioned!*
Added

*And then, please provide mass concentrations instead of volume concentrations in Fig. 17.*

We changed Fig.17 and added profiles of mass concentration. Corresponding paragraph is also added to the text.

*Acknowledgement: A statement concerning ACTRIS is missing, but required to my opinion.*
Added

*Fig. 2, caption: 350-2800 m.*
Done

Fig. 12, caption: Maybe in line 589: … depict the total particle volume….
Done

*General remark to the figures: There should be always (a) (b) (c) when there are several panels. Sometimes it is written (a) ... and (b) ... in the caption, but no indication of panels in terms of (a) and (b).*

Done

---

## Author Comment (AC2)

**Response to Referee 2**

We would like to thank the Referee for reading the manuscript and for kind words about our work.

*The work is devoted to the problem of recognizing and identifying the type of aerosol in lidar measurements. It is known that aerosols in the atmosphere arise from numerous sources and undergo intense mixing during atmospheric transport. Typically, the aerosol environment at a particular geographic location is a mixture of different types of particles. In this regard, the task of recognizing aerosol particles by their origin is extremely important. The article under discussion presents an original approach to recognizing the components of aerosol mixtures based on the analysis of polarization and spectroscopic properties of backscattering coefficients. The proposed method uses data obtained with the Mi-Raman fluorescent lidar at the University of Lille, France. This lidar is well known from the authors' past publications and, with its advanced measurement capabilities, provides rich information on aerosols.*

*The authors propose to take the aerosol backscattering coefficient, the particle depolarization coefficient, and the fluorescent ability as the main input parameters to identify the type of aerosol. Fluorescence ability is defined as the ratio of the fluorescence backscatter coefficient to the backscatter coefficient of particles at the un shifted scattering line. In this case, it is possible to write a system of three lidar equations, the solution of which allows one to determine the composition of the three-component aerosol mixture. The work shows that the method allows one to reliably separate the contributions of such types of aerosols as forest fire smoke, urban aerosol, and dust into the total backscattering coefficient at a wavelength of 532 nm. To achieve this, much work has been done on statistical analysis of the scattering properties of each of these types of aerosols. Obviously, due to the variability of the scattering properties of aerosols, only some statistical relationship is possible between the type of aerosol and the combination of its scattering properties. Understanding this well, the authors use the Monte Carlo method, provided that the scattering properties of aerosols are uniformly distributed within predetermined intervals. This approach provides a statistical sample of possible solutions, allowing the average and dispersion to be calculated over the entire set of solutions. In this case, the dispersion serves as an indicator of the uncertainty of the method. This approach was tested using 28 observations carried out at the ATOLL Observatory, Laboratoire d'Optique Atmosphérique, University of Lille, France.*

*The presented publication is undoubtedly an important step towards improving lidar methods for their more complete use for studying the dynamics of aerosol layers, atmospheric circulation processes, global transport phenomena, etc. The proposed approach can significantly expand the capabilities of lidar systems and increase the efficiency of lidar networks.*
*It should be noted that the proposed method has some limitations. There is no doubt that the method works well for dry aerosol and it is quite obvious that it will stop working in case of wetting of aerosol particles. Fortunately, the authors, being experienced experimenters, understand this and discard measurements made in conditions of relative humidity greater than 60%.*

*However, it is not very clear from the publication how they control the relative humidity along the sounding path. This suggests some recommendation to the authors to supplement their lidar, for example, with Raman channels for measuring temperature and humidity. This modernization is quite possible. In this case, all questions about the applicability of the method would be automatically resolved.*

To derive the RH profile we use water vapor measurements of our lidar and the temperature profile from radiosonde in Belgium. The measurements are not collocated, but this is the best from

available at the moment. The Reviewer is right; temperature measurements would be very useful to increase capability of our instrument. Incorporation of temperature channel is in our plans.

*According to formal characteristics, the article fully complies with the requirements of the AMT, since it discusses scientific issues relevant to the topic of AMT. The publication presents new concepts, ideas, tools, and data. Substantial conclusions are drawn and the scientific methods and assumptions are reasonable and clearly stated. The findings are sufficient to support interpretations and conclusions. Descriptions of experiments and calculations are presented quite fully and accurately, and can be reproduced by fellow scientists. The authors pay due attention to the achievements of colleagues in the field under discussion and clearly indicate their original contributions. The title of the article reflects the content. The abstract contains a brief and complete summary of the content of the article. The overall composition of the article is well structured and understandable. The language of presentation is quite free and precise. Mathematical formulas, symbols, abbreviations, and units of measurement are defined correctly and used as intended. There is no need to shorten, combine or exclude any parts of the article (text, formulas, figures, tables). The number and quality of links correspond to the content of the text. The quantity and quality of additional material meets the requirements. To summarize, it should be noted that the publication under discussion makes the most pleasant impression, is a significant contribution to the development of remote methods for monitoring atmospheric parameters, and can be published without modifications.*

Thanks again for high assessment of our work.

---

## Author Comment (AC3)

**Response to Referee 3**

We would like to thank the Reviewer for reading the manuscript and for useful suggestions. Below, we provide response to the comments.

*The authors have developed the partitioning method of smoke, urban, and dust aerosols based on Mie-Raman-fluorescence lidar measurements and have shown excellent performance. Classification of aerosol types and quantification of their respective components is very important in atmospheric environment and climate change. In particular, the partitioning of smoke and urban aerosols is a significant contribution to remote sensing methods. The methods, results, and suggestions are reasonable and clearly described. I recommend that this paper can be published with some minor modifications.*

*Specific comments*

*Lines 143-144: How did you introduce the non-negativity constraint to the least squares method?*

The non-negativity constraint was implemented as follows. First, the LSQ problem was being solved in 3D space without non-negativity restrictions. If the solution was non-negative, it was taken for the final result. Otherwise, the LSQ problem was being solved on three 2D planes $(\eta_s=0)$, $(\eta_d=0)$, $(\eta_u=0)$. If non-negative solution(s) were found, one of them having the least discrepancy was taken for the final result. Otherwise, the process was repeated for 1D lines $(\eta_s=0, \eta_d=0)$, $(\eta_s=0, \eta_u=0)$, $(\eta_d=0, \eta_u=0)$. If no non-negative solution(s) were found on this last stage, the final solution was (0,0,0).

However, we would not like to put all these details in the manuscript. It will look to "mathematical". In the revised manuscript, we tried to simplify description of LSQ solving, to make it easier for reader.

*Lines 154-156 and 162-163: The partitioning method would be helpful for atmospheric environment monitoring and data assimilation. The calculations of the ATS method for the four triplets seem time consuming. Is the method applicable to the quasi-real-time analysis?*

Yes, the ATS method is time consuming. To analyze the night measurement session (Fig.9) it takes about 40 min for standard notebook computer. However, when only 3 aerosol types are considered, computation time is about 8 min, so quasi-real-time analysis is possible. We do not provide these numbers in the manuscript, because the computation time depends on the parameters of computer used.

*Lines 177-129: What are the ranges of fluorescence capacities and depolarization ratios for smoke, pollen, urban, and dust aerosols above 60 % relative humidity? If several studies exist, their ranges should be noted for reference.*

Decrease of the fluorescence capacity and the particle depolarization ratio in the process of hygroscopic growth was demonstrated in recent publication of Veselovskii et al. (2024) in Fig.6. The $G_F$ decreases from $1.2\times10^{-4}$ to $0.1\times10^{-4}$, while $\delta_{532}$ from 9% to 3% when RH increases up to 90%. We should mention, that the hygroscopic growth does not affect the spectrum of fluorescence. Therefore, the use of two or more fluorescence channels allows particle identification even at high RH. The comment and reference is added to the manuscript.

*Table 1: Why is the fluorescence capacities of smoke and pollen so large? A brief explanation is in the best interest of the reader.*

High fluorescence capacity of smoke is due to the presence of organic carbon. Biological materials are responsible for strong fluorescence of pollen. Added to the text.